Fishery catch is affected by geographic expansion, fishing down food webs and climate change in Aotearoa, New Zealand

http://orcid.org/0000-0002-2068-1080 Lavin Charles Patrick 1 charles.p.lavin@nord.no
Pauly Daniel 2
http://orcid.org/0000-0003-3412-3503 Dimarchopoulou Donna 3 4
Liang Cui 5
http://orcid.org/0000-0003-2362-0328 Costello Mark John 1
1 Faculty of Bioscience and Aquaculture, Nord University , Bodø , Norway
2 Sea Around Us, Institute for the Ocean and Fisheries, University of British Columbia , Vancouver, British Columbia , Canada
3 Biology Department, Dalhousie University , Halifax, Nova Scotia , Canada
4 Biology Department, Woods Hole Oceanographic Institution , Woods Hole, Massachusetts , United States
5 Key Laboratory of Marine Ecology and Environmental Science, Institute of Oceanology, Chinese Academy of Sciences , Qingdao , China
Aguilar-Perera Alfonso
Electronic publication date: 2023 Sep 21
Publication date: 2023
Volume: 11
Electronic Location ID: e16070
Received 2023 May 8; Accepted 2023 Aug 20
Copyright: © 2023 Lavin et al.
Copyright year: 2023
Copyright holder: Lavin et al.
License: This is an open access article distributed under the terms of the Creative Commons Attribution License, which permits unrestricted use, distribution, reproduction and adaptation in any medium and for any purpose provided that it is properly attributed. For attribution, the original author(s), title, publication source (PeerJ) and either DOI or URL of the article must be cited.
License URL: https://creativecommons.org/licenses/by/4.0/

Keywords: New Zealand, Fisheries, Ocean warming, Mean trophic level, Mean temperature of the catch, Fishing down marine food webs, Fishing-in-Balance index

Funding: The authors received no funding for this work.

==============================
Historical fishing effort has resulted, in many parts of the ocean, in increasing catches of smaller, lower trophic level species once larger higher trophic level species have been depleted. Concurrently, changes in the geographic distribution of marine species have been observed as species track their thermal affinity in line with ocean warming. However, geographic shifts in fisheries, including to deeper waters, may conceal the phenomenon of fishing down the food web and effects of climate warming on fish stocks. Fisheries-catch weighted metrics such as the Mean Trophic Level (MTL) and Mean Temperature of the Catch (MTC) are used to investigate these phenomena, although apparent trends of these metrics can be masked by the aforementioned geographic expansion and deepening of fisheries catch across large areas and time periods. We investigated instances of both fishing down trophic levels and climate-driven changes in the geographic distribution of fished species in New Zealand waters from 1950–2019, using the MTL and MTC. Thereafter, we corrected for the masking effect of the geographic expansion of fisheries within these indices by using the Fishing-in-Balance (FiB) index and the adapted Mean Trophic Level (aMTL) index. Our results document the offshore expansion of fisheries across the New Zealand Exclusive Economic Zone (EEZ) from 1950–2019, as well as the pervasiveness of fishing down within nearshore fishing stock assemblages. We also revealed the warming of the MTC for pelagic-associated fisheries, trends that were otherwise masked by the depth- and geographic expansion of New Zealand fisheries across the study period.

Introduction

Shifts in marine species’ geographic distribution with ocean warming are increasingly well documented (Cheung, Watson & Pauly, 2013; Pinsky, Selden & Kitchel, 2020; Chaudhary et al., 2021). Further shifts may alter community structure (García Molinos et al., 2016; Poloczanska et al., 2016), food web interactions (Tekwa, Watson & Pinsky, 2022), and fisheries catch (Cheung et al., 2010; Cheung, Watson & Pauly, 2013). Ocean warming can also impact the fitness of marine ectotherms, particularly large, active predatory species (Forster, Hirst & Atkinson, 2012; van Rijn et al., 2017; Lavin et al., 2022). Concurrently, historical fishing pressure has resulted in deleterious impacts on marine communities and ecosystems (Howarth et al., 2014). Prolonged, intensive fishing results in reductions of population biomass (Worm et al., 2009), as well as the maximum size of harvested species, as fisheries preferentially target the largest species and individuals within a population (Jackson et al., 2001) due to their higher market price (Tsikliras & Polymeros, 2014). Once the largest species, often of the highest trophic levels and long-lived, have been depleted, fishing pressure has been observed to shift to smaller, shorter-lived species of lower trophic levels—termed ‘fishing down marine food webs’ (Pauly et al., 1998; Pauly & Palomares, 2005). The occurrence of fishing down can have deleterious consequences to the transfer of energy through marine food webs, thus impacting marine biodiversity and ecosystem function though the depletion of high trophic level predators (Pauly et al., 1998; Baum & Worm, 2009; Boyce et al., 2015). Thus, both fishing impacts and climate-change related outcomes to marine populations should be considered in concert in order to inform fisheries and conservation management (Lynam et al., 2017).

One method to infer changes in the geographic distribution of marine fish and invertebrate populations includes the Mean Temperature of the Catch index (MTC, Cheung, Watson & Pauly, 2013). This index represents the catch-weighted mean temperature preference of species in the fisheries catch in an area, and has been applied to a variety of marine ecosystems globally (see Leitão et al., 2018 and Dimarchopoulou et al., 2021b and references therein). Overall, an increase in MTC over time indicates an ocean-warming induced increase in the proportion of species of warmer thermal affinity relatively to those of colder thermal affinity recorded in the catch (Cheung, Watson & Pauly, 2013; Tsikliras & Stergiou, 2014; Liang, Xian & Pauly, 2018). Similarly, one method to infer the occurrence of fishing down includes the Mean Trophic Level index (MTL, also known as the Marine Trophic Index) (Pauly et al., 1998; Pauly & Watson, 2005), i.e., the catch-weighted mean trophic level of fisheries within an area. Also utilized extensively (see Liang & Pauly, 2017, and references therein), a decrease in MTL indicates the transition of catch from large, long-lived, higher trophic level species to smaller, short-lived, lower trophic level species, as a result of overexploitation and thus fishing down (Pauly et al., 1998). Conversely, increasing trends in MTL indicate the increasing catch of higher trophic level species, as has been observed during the historical expansion of fisheries (Leitão, 2015).

Yet, for both of these well-utilized indices, apparent trends in catch-weighted values can be masked by a variety of factors. One masking factor includes the ‘skipper effect,’ in that skippers will continuously and preferentially target certain species, usually of high trophic levels, within a multispecies catch (Pinnegar et al., 2002; Liang & Pauly, 2020). This may also arise when catch quotas prevent shifting catch from one species to another. Another masking effect includes the taxonomic resolution at which analyses are performed, as trends in catch-weighted values may not be apparent when analyzed at coarse taxonomic resolutions (Pauly & Palomares, 2005; Liang & Pauly, 2017). Lastly, when such indices are calculated across large geographic areas (e.g., exclusive economic zones, EEZs) and time scales, the gradual geographic and depth expansion of fisheries to include new habitats and species (i.e., deep-water species of colder thermal affinity, or higher trophic level species) can mask trends of fishing down or MTC warming (Bhathal & Pauly, 2008; Kleisner, Mansour & Pauly, 2014).

Fisheries in Aotearoa (henceforth referred to as New Zealand) waters have experienced a significant transformation within the 20th century through technological proliferation and government incentives. Following the establishment of their EEZ in 1978, New Zealand became the first country to implement a Quota Management System (QMS) in 1986, in recognition of declining stocks of important coastal fisheries (Johnson & Haworth, 2004). Since then, New Zealand has prioritized the transition towards ecosystem-based fisheries management in line with the targets outlined in the Convention on Biological Diversity (Cryer, Mace & Sullivan, 2016; Durante, Beentjes & Wing, 2020). Overall, the transition towards an ecosystem-based, multi-species management scheme requires in-depth understanding of the historical effects of fisheries (Pauly, 1995), as well as those effects presented by trends in ocean warming (Free et al., 2019) in order to inform management under multiple and interacting stressors in a changing ocean.

To investigate the historical impacts of fisheries expansion in New Zealand waters, Durante, Beentjes & Wing (2020) calculated the MTL using New Zealand fisheries catch data from 1930–2014. The authors found a positive trend in MTL coinciding with a period of geographic expansion of the fisheries, followed by a negative trend in MTL, thus displaying signals of fishing down. Yet, the authors did not correct the MTL trend for the masking effects of geographic expansion of fisheries across the time series (Durante, Beentjes & Wing, 2020). In the present study, we do so by applying the Fishing-in-Balance (FiB) index (Pauly et al., 1998; Pauly, Christensen & Walters, 2000; Pauly & Watson, 2005; Bhathal & Pauly, 2008), and the adapted Mean Trophic Level index (aMTL, also known as the region-based Marine Trophic Index, Kleisner, Mansour & Pauly, 2014; Liang & Pauly, 2017) to New Zealand fisheries catch data from 1950–2019. By accounting for the geographic expansion of fisheries in New Zealand waters, we investigated whether fishing down occurred concurrent with fishery expansion. Further, by separating catch data between species’ habitat and gear used, we also investigated whether signals of warming in the MTC were apparent across the study period in line with trends in ocean warming.

Materials and Methods

The study area was the New Zealand EEZ (Fig. 1). To quantify the rate of ocean warming, we extracted Sea Surface Temperature Anomaly (SSTA) data from NOAA’s Kaplan Extended SST V2 database (Kaplan et al., 1998). At 5° latitude by 5° longitude resolution, monthly values were averaged across years from 1950–2019. These SSTA data are relative to temperatures during the period from 1951–1980 (Kaplan et al., 1998), and SSTA was chosen as temperature anomalies are considered to present a more consistent indicator of climate-change related ocean warming across large areas relative to absolute Sea Surface Temperatures (Tsikliras & Stergiou, 2014; Tsikliras et al., 2015; Dimarchopoulou et al., 2021b). We then performed simple linear regression and segmented linear regression of SSTA on time (Years). Segmented linear regressions were performed in order to test whether trends were better explained by multiple regression lines (Dimarchopoulou et al., 2021b), as indicated by the presence of a significant breakpoint (Muggeo, 2016). Breakpoints were identified using the segmented package in R (Muggeo, 2008; Team R Development Core, 2022), whereby a non-zero difference in slope on either side of the breakpoint was tested using a p-score test from the segmented package (Muggeo, 2008).

Figure 1 The study area.

Fisheries catch data were extracted for the New Zealand Exclusive Economic Zone (EEZ, black line). Also shown is the 12 nautical mile Territorial Sea border (red), as well as depth (m). Map source credit: GADM.

Primary analyses were completed using fisheries catch data for the New Zealand EEZ extracted from the Sea Around Us database (Pauly, Zeller & Palomares, 2020). This database includes reconstructed catches of officially reported statistics from the Food and Agriculture Organization of the United Nations (FAO), plus estimated unreported catch, discards and by-catch from all fishing sectors operating in the area, from 1950–2019 (Simmons et al., 2016; Pauly, Zeller & Palomares, 2020, Fig. 2A). In order to complete MTC and MTL analyses, species traits’, including mean preferred temperature and trophic level were gathered from FishBase (Froese & Pauly, 2021) via the rfishbase package in R (Boettiger, Lang & Wainwright, 2012; Team R Development Core, 2022). The mean preferred temperature (°C) for each species is based on a compilation of modelled species’ distributions informed by occurrence data, and also adjusted for appropriate depth ranges (Cheung, Watson & Pauly, 2013; Froese & Pauly, 2021). A species’ trophic level (TL) is estimated via diet information in FishBase (Froese & Pauly, 2021), and is represented by a discrete value usually between 2.0 and 5.0 (Liang & Pauly, 2017). The trophic level represents the number of trophic links a species maintains from primary production (TL = 1), i.e., herbivorous fish have a trophic level = 2, piscivorous fish have a trophic level = 3, and so on, with the maximum trophic levels in marine systems exceeding a value of five (i.e., top predators) (Stergiou & Karpouzi, 2002; Dimarchopoulou et al., 2021a; Eddy et al., 2021).

Figure 2 Fisheries catch data analyzed.

Fisheries catch (solid areas, tons × 106, left axis) including (A) the full catch dataset extracted from the Sea Around Us database, (B) the restricted Sea Around Us catch dataset that was analyzed in the present study, and (C) catch data analyzed in sensitivity analyses. The full dataset (A) included catch information at the species level (black line, right axis) and also for groups at higher taxonomic resolution (red line, right axis), while the catch data analyzed (B) was restricted to the species level (black line, right axis), composed of catch from species both included and not included in New Zealand’s Quota Management System (QMS), from reported and non-reported sources (i.e., reconstructed). (C) FAO catch data of n = 42 species analyzed in sensitivity analyses (black area, left axis), compared to the analyzed Sea Around Us catch data (n = 102, solid black line, left axis), and the further reduced Sea Around Us catch data (n = 42, dashed line, left axis) also contained in sensitivity analyses.

The number of species with available trait information contained in the catch data was 110. After identifying these species, we compared their Sea Around Us catch trends to officially reported statistics from the FAO (FAO, 2020). For several taxonomically-similar species, or species that were historically mislabeled, catch statistics have been reported under a group code or at higher taxonomic resolution (ex. Allocyttus niger and Pseudocyttus maculatus combined as ‘Oreo species,’ Simmons et al., 2016). In the present study, we restricted analyses to the species-level, as applications of the MTC and MTL at higher taxonomic coarseness have been demonstrated to mask the effects of fishing down trophic levels (Pauly & Palomares, 2005; Liang & Pauly, 2017). We therefore removed from analyses species that were previously reported in higher taxonomic groups (n = 7), were without available trait information (n = 19), and records labelled at a higher or an unidentified taxonomic resolution (n = 103) (Table S1).

Following this data treatment 102 species were included in analyses (Table 1). They represented 80% of the total species recorded and 48% of the total raw catch data contained in New Zealand’s Sea Around Us dataset (Fig. 2B). The majority of this analyzed catch data is of QMS species (96% of data), 67% of which is reported catch and 29% of which is unreported catch (i.e., reconstructed, Fig. 2B). We retained several ‘rare’ species (of low catch quantities) or bycatch-related species in our analyses. This was done as fish community composition can change over time, such as due to climate warming and shifts in species’ geographic distribution to new areas (Gordó-Vilaseca et al., 2023). We then calculated the MTC of New Zealand’s catch using the formula:

Table 1 Study species information.

Species	Milieu	Mean pref. temp. (°C)	Trophic level	Mean annual catch (tons) ± SE	(log) Fisheries catch slope across time series	QMS species	FAO species	
Beryx decadactylus	Bathydemersal	11.2	4.13	0.3 ± 0.3	0.50	Yes		
Capromimus abbreviatus	Bathydemersal	13	3.45	11.9 ± 3.4	0.13			
Centriscops humerosus	Bathydemersal	7.3	3.57	26.7 ± 3.1	0.08			
Centrophorus squamosus	Bathydemersal	7	4.47	8.8 ± 1.7	0.09			
Centroselachus crepidater	Bathydemersal	7.8	4.16	1.7 ± 0.7	0.03			
Cyttus traversi	Bathydemersal	8.3	3.93	505.4 ± 36.2	0.03	Yes	Yes	
Dalatias licha	Bathydemersal	5.3	4.23	315.7 ± 23.6	−0.03			
Diastobranchus capensis	Bathydemersal	7	4.50	6.0 ± 2.0	0.20			
Epigonus telescopus	Bathydemersal	8.9	3.59	2,106.4 ± 243.9	−0.02	Yes	Yes	
Eptatretus cirrhatus	Bathydemersal	12.3	5.00	166.4 ± 48.8	0.33			
Genypterus blacodes	Bathydemersal	7.2	4.18	10,674.8 ± 1,074.9	0.07	Yes	Yes	
Harriotta raleighana	Bathydemersal	5.2	3.55	78.3 ± 10.9	0.20			
Hydrolagus novaezealandiae	Bathydemersal	10.8	3.52	1,831.3 ± 117.4	0.04	Yes	Yes	
Macrourus carinatus	Bathydemersal	6.7	3.70	19.1 ± 6.4	0.61			
Pentaceros decacanthus	Bathydemersal	11.4	3.42	1.7 ± 1.4	0.13			
Plagiogeneion rubiginosum	Bathydemersal	14.2	3.40	449.1 ± 36.1	0.04	Yes	Yes	
Tripterophycis gilchristi	Bathydemersal	10.3	3.03	6.5 ± 6.5	−0.10			
Alepisaurus ferox	Bathypelagic	8.9	4.03	0.2 ± NA	NA			
Antimora rostrata	Bathypelagic	2.9	3.58	5.4 ± 1.0	0.14			
Centrolophus niger	Bathypelagic	7	3.92	46.4 ± 5.9	0.02			
Halargyreus johnsonii	Bathypelagic	3.7	3.38	4.3 ± 1.0	0.09			
Hoplostethus atlanticus	Bathypelagic	6.6	4.25	30,114.6 ± 3,679.4	−0.03	Yes	Yes	
Lampris guttatus	Bathypelagic	12.1	4.22	133.6 ± 19.8	0.01	Yes	Yes	
Lepidorhynchus denticulatus	Bathypelagic	9.2	3.67	3,054.8 ± 337.5	0.09			
Magnisudis prionosa	Bathypelagic	6.8	4.50	0.7 ± 0.3	−0.08			
Mora moro	Bathypelagic	5	3.75	840.5 ± 79.3	0.06	Yes	Yes	
Neocyttus rhomboidalis	Bathypelagic	7.8	3.58	121.9 ± 9.5	0.13	Yes	Yes	
Paratrachichthys trailli	Bathypelagic	13.3	3.50	7.8 ± 1.1	0.13			
Trachipterus trachypterus	Bathypelagic	13.5	2.88	46.3 ± 10.9	−0.08			
Beryx splendens	Benthopelagic	9.2	4.27	2,652.6 ± 109.4	0.02	Yes		
Centroberyx affinis	Benthopelagic	15.9	3.81	80.2 ± 8.1	0.02	Yes	Yes	
Galeocerdo cuvier	Benthopelagic	26.4	4.56	0.3 ± 0.1	0.06			
Galeorhinus galeus	Benthopelagic	12.3	4.34	2,364.7 ± 203.4	0.04	Yes	Yes	
Girella tricuspidata	Benthopelagic	16.6	2.09	77.4 ± 4.3	0.01	Yes	Yes	
Hyperoglyphe antarctica	Benthopelagic	7.5	3.95	1,708.8 ± 161.9	0.11	Yes	Yes	
Lepidocybium flavobrunneum	Benthopelagic	9.9	4.34	31.3 ± 4.9	0.10			
Lepidopus caudatus	Benthopelagic	12.1	3.82	2,474.8 ± 186.0	0.07	Yes	Yes	
Macruronus novaezelandiae	Benthopelagic	8	4.53	115,806.5 ± 12,799.5	0.13	Yes	Yes	
Merluccius australis	Benthopelagic	8.6	4.28	7,110.5 ± 854.3	0.16	Yes	Yes	
Micromesistius australis	Benthopelagic	5.8	3.66	38,893.7 ± 3,142.3	−0.01	Yes	Yes	
Mugil cephalus	Benthopelagic	23.2	2.48	773.6 ± 40.5	0.02	Yes		
Odontaspis ferox	Benthopelagic	17.3	4.16	0.3 ± 0.1	−0.23			
Ruvettus pretiosus	Benthopelagic	12.9	4.18	41.0 ± 6.6	−0.05			
Scorpis violacea	Benthopelagic	17.9	3.44	19.2 ± 7.3	0.02			
Seriola lalandi	Benthopelagic	14.9	4.16	913.9 ± 27.5	0.01	Yes		
Seriolella brama	Benthopelagic	14.4	3.73	2,505.5 ± 240.8	0.06	Yes	Yes	
Seriolella punctata	Benthopelagic	13	3.53	10,172.2 ± 397.1	0.00	Yes	Yes	
Sphoeroides pachygaster	Benthopelagic	19.4	4.20	0.5 ± 0.2	−0.28			
Squalus acanthias	Benthopelagic	9.9	4.37	6,191.8 ± 422.2	0.04	Yes	Yes	
Squalus mitsukurii	Benthopelagic	14.1	4.37	76.1 ± 6.6	−0.04			
Thyrsites atun	Benthopelagic	11.2	3.63	23,589.3 ± 1,776.3	0.07	Yes	Yes	
Zenopsis nebulosa	Benthopelagic	14.5	4.39	77.4 ± 9.9	0.16			
Aldrichetta forsteri	Demersal	17.1	2.51	39.8 ± 2.7	−0.01	Yes	Yes	
Argentina elongata	Demersal	11.2	3.40	70.4 ± 9.7	0.03			
Callorhinchus milii	Demersal	15.2	3.60	1,428.2 ± 63.3	0.00	Yes	Yes	
Chelidonichthys kumu	Demersal	19.3	3.68	4,532.9 ± 144.8	0.00	Yes	Yes	
Cyttus novaezealandiae	Demersal	13.1	3.67	242.3 ± 47.4	−0.02			
Genyagnus monopterygius	Demersal	16.6	4.50	42.1 ± 2.0	−0.02			
Kathetostoma giganteum	Demersal	11.2	4.21	1,950.4 ± 189.4	0.09	Yes	Yes	
Meuschenia scaber	Demersal	15.1	2.83	383.3 ± 41.1	0.07	Yes		
Mustelus lenticulatus	Demersal	13.7	3.51	2,051.1 ± 148.3	0.01	Yes	Yes	
Notopogon lilliei	Demersal	11.7	3.52	6.3 ± 2.0	−0.04			
Notorynchus cepedianus	Demersal	14.2	4.68	9.1 ± 1.7	0.06			
Parapercis colias	Demersal	12.6	3.89	2,452.1 ± 97.0	0.00	Yes	Yes	
Paristiopterus labiosus	Demersal	15.9	3.33	18.1 ± 4.5	−0.03			
Pseudophycis bachus	Demersal	11	3.93	7,111.3 ± 742.2	0.04	Yes	Yes	
Arripis trutta	Pelagic-neritic	17.4	4.07	987.5 ± 19.7	0.00	Yes	Yes	
Brama brama	Pelagic-neritic	11.8	4.08	364.6 ± 44.9	0.00	Yes	Yes	
Coryphaena hippurus	Pelagic-neritic	27.4	4.21	82.5 ± 13.4	0.02			
Sardinops sagax	Pelagic-neritic	17.9	2.84	318.8 ± 54.9	0.08	Yes	Yes	
Scomber australasicus	Pelagic-neritic	18.7	4.23	8,201.8 ± 790.6	0.09	Yes	Yes	
Seriolella caerulea	Pelagic-neritic	9.2	3.20	1,582.1 ± 151.3	0.06	Yes	Yes	
Allothunnus fallai	Pelagic-oceanic	18.4	3.70	16.3 ± 2.6	0.07			
Alopias superciliosus	Pelagic-oceanic	27.1	4.46	0.5 ± 0.2	−0.34			
Alopias vulpinus	Pelagic-oceanic	23.3	4.50	18.9 ± 1.8	0.03			
Carcharhinus longimanus	Pelagic-oceanic	26.8	4.16	9.9 ± 1.0	−0.03			
Carcharodon carcharias	Pelagic-oceanic	18.1	4.53	1.1 ± 0.4	0.02			
Cetorhinus maximus	Pelagic-oceanic	11.4	3.20	960.1 ± 92.6	0.05			
Dissostichus eleginoides	Pelagic-oceanic	4.5	4.49	11.7 ± 2.5	0.23	Yes		
Istiompax indica	Pelagic-oceanic	25.4	4.50	1.5 ± 0.2	0.02			
Istiophorus platypterus	Pelagic-oceanic	25.6	4.50	0.1 ± 0.1	−0.26			
Isurus oxyrinchus	Pelagic-oceanic	17.4	4.52	54.7 ± 7.6	0.04	Yes	Yes	
Kajikia audax	Pelagic-oceanic	25.9	4.50	95.9 ± 11.5	0.02			
Katsuwonus pelamis	Pelagic-oceanic	26.2	4.43	4,919.9 ± 644.4	0.09			
Lamna nasus	Pelagic-oceanic	7.8	4.46	80.0 ± 11.6	0.08	Yes	Yes	
Makaira Mazara	Pelagic-oceanic	19.3	4.46	0.1 ± 0.0	−0.01			
Mola mola	Pelagic-oceanic	10.2	3.28	1.2 ± 0.2	0.00			
Prionace glauca	Pelagic-oceanic	14.8	4.35	886.8 ± 76.4	0.07	Yes	Yes	
Regalecus glesne	Pelagic-oceanic	23.9	3.20	9.5 ± 4.6	−0.33			
Sphyrna zygaena	Pelagic-oceanic	26.5	4.94	12.2 ± 0.6	0.00			
Tetrapturus angustirostris	Pelagic-oceanic	26.2	4.50	0.2 ± 0.2	0.01			
Thunnus alalunga	Pelagic-oceanic	15.1	4.30	1,729.0 ± 147.0	0.03			
Thunnus albacares	Pelagic-oceanic	26.7	4.41	197.4 ± 29.2	−0.03	Yes	Yes	
Thunnus maccoyii	Pelagic-oceanic	5	3.93	1,959.7 ± 318.8	0.00	Yes	Yes	
Thunnus obesus	Pelagic-oceanic	26.6	4.42	384.9 ± 35.9	0.01	Yes	Yes	
Thunnus orientalis	Pelagic-oceanic	24.3	4.50	107.8 ± 14.4	0.07	Yes	Yes	
Xiphias gladius	Pelagic-oceanic	22.7	4.53	532.3 ± 49.0	0.05	Yes	Yes	
Carcharhinus brachyurus	Reef-associated	17.4	4.33	23.1 ± 2.0	−0.03			
Carcharhinus falciformis	Reef-associated	26.7	4.35	70.2 ± 7.1	−0.01			
Carcharhinus galapagensis	Reef-associated	23.8	4.23	0.1 ± NA	NA			
Latris lineata	Reef-associated	14.9	3.41	45.3 ± 5.4	0.05	Yes		
Pagrus auratus	Reef-associated	17.4	3.59	16,163.0 ± 930.4	−0.01	Yes	Yes	
Note:

Species trait information included in mean temperature of the catch (MTC) and mean trophic level (MTL) calculations (n = 102). Included are the species milieu (i.e., habitat association), species’ mean preferred temperature (°C), trophic level, mean annual catch (tons ± SE), and the slope of (log) catch values across the time series. Species’ milieu classifications were gathered directly from the FishBase online database (Froese & Pauly, 2021), while mean temperature preference and trophic level were accessed via the rfishbase package in R (Boettiger, Lang & Wainwright, 2012; Team R Development Core, 2022). Fisheries catch data were extracted for the New Zealand EEZ from the Sea Around Us (Pauly, Zeller & Palomares, 2020). We also list which species are included in New Zealand’s Quota Management System (QMS), as well, which species are contained within the FAO dataset (n = 42) analyzed for sensitivity analyses (FAO, 2020). The same FAO species are also included within the reduced Sea Around Us dataset (n = 42) contained in further sensitivity analyses.

(1) MTCyr=∑in⁡TiCi,yr∑in⁡Ci,yr

where Ti is the preferred mean temperature of species i, Ci,yr is the catch of species i in year yr, and n equals to total number of species recorded (Cheung, Watson & Pauly, 2013). We then calculated the MTL of the same catch data, using the formula:

(2) MTLyr=∑im⁡TLiYik∑im⁡Yik

where TLi is the trophic level of species i, Yik is the catch of species i in year k, and m is the number of species recorded (Pauly et al., 1998). For both MTC and MTL trends, we calculated simple and segmented linear regressions of each index on time (Year) across the study period. In order to relate MTC trends to trends in ocean warming, we computed Kendall’s τ correlation between MTC and SSTA using the stats package in R (Team R Development Core, 2022); with the null hypothesis that there is no correlation between the two variables. Thereafter, in order to first address the geographic expansion of fisheries and thus the masking of MTC and MTL trends, we calculated the Fishing-in-Balance (FiB) index. The FiB index is designed to identify surplus fisheries catch (i.e., from adjacent areas or stock assemblages) based on assumptions of energy transfer between trophic levels. The FiB was calculated using the formula:

(3) FiBk=log10[Yk(1TE)MTLk]−log10[Y0(1TE)MTL0]

where Y is the catch and MTL is the mean trophic index value for year k, while Y0 and MTL0 are the catch and MTL value for the first year of data, and TE is the transfer efficiency between trophic levels (TE = 0.1), as estimated and assumed constant by Pauly & Christensen (1995) and used in Bhathal & Pauly (2008). Under this assumption of trophic transfer efficiency, a decrease in TL of 1 should yield a 10-fold increase in catch, or conversely, an increase in TL of 1 should yield a 10-fold decrease in catch (Kleisner, Mansour & Pauly, 2014; Liang & Pauly, 2017). As such, fisheries are considered to be “fishing in balance” (FiB = 0) when a decline in MTL value corresponds to the expected increase in catch, or vice versa, in line with assumptions of constant TE (Kleisner, Mansour & Pauly, 2014). However, when catch increases more than what is expected from decreases in MTL (FiB > 0), this implies that excess catch came from adjacent stocks and thus indicates the geographic expansion of fisheries (Kleisner, Mansour & Pauly, 2014; Liang & Pauly, 2017; Dimarchopoulou et al., 2021a). We also applied linear and segmented regression to the trend of FiB across the study period (Muggeo, 2008).

In order to quantify the geographic expansion of fisheries across the time series, we then computed the adapted Mean Trophic Level (aMTL) in line with Kleisner, Mansour & Pauly (2014) and Liang & Pauly (2017). The aMTL was developed under the assumption that geographic expansion occurs once a certain fished area or stock shows signs of depletion. Considering costs of fuel and equipment, it is assumed that most fishing activities are first concentrated within nearshore waters. Once the nearshore fisheries become overexploited, fishing effort then expands further away from shore and/or deeper (Kleisner, Mansour & Pauly, 2014; Liang & Pauly, 2017). Based on the FiB index (Eq. (3)), if fishing occurs “in balance”, where FiB = 0, with initial catch Y0 and MTL0, catch Yk of year k can be computed:

(4) Yk=Y0×(1TE)MTL0−MTLk

As such, if Yk>Y0×(1TE)MTL0−MTLk, this indicates that a geographic expansion of the fisheries has occurred (Liang & Pauly, 2017), i.e., accessing previously unexploited assemblage of fish stocks in adjacent areas, habitats or depths (Kleisner, Mansour & Pauly, 2014). Upon identification of an expansion, a ‘node’ marks the year of occurrence, after which new aMTL values can be calculated for those newly identified stock assemblages (Kleisner, Mansour & Pauly, 2014). This identification of expansion operates on two main assumptions: that fishing does not cease in the original stock assemblages, and that fishing in the original stock assemblage is in balance (i.e., FiB = 0), or in decline (Kleisner, Mansour & Pauly, 2014).

Since the initial mean trophic level (MTL0) in Eq. (4) may not be fully representative of all trophic levels present within the ecosystem, we corrected for this by assigning a range of possible trophic levels between the lowest TL (TLlower) and highest TL (TLupper) within the catch data (Liang & Pauly, 2017). Amongst J trophic levels, within the range [TLlower, TLupper], at each trophic level j, the catch potential p Ykj was calculated:

(5) pYkj=Y0×(1TE)TLj−MTLk

while the corresponding maximum catch potential for year k is calculated:

(6) pYkj=∑j=1J⁡(pYkj×Pr(TLj))

with Pr( TLj) equal to the probability that MTL0 = TLj (Liang & Pauly, 2017). By calculating p Ykj, which is independent of MTL0, we are able to estimate the maximum catch value that fisheries could achieve within a distinct stock assemblage, under the assumption of TE. When a reported catch Yk > p Yk, this indicates a geographic expansion has occurred, whereby year k is assigned node nr, and r indicates the newly identified stock assemblage (Liang & Pauly, 2017). For any year and stock assemblage that follows nr, catch and aMTL was calculated amongst distinct assemblages, with the maximum number of assemblages computed = 3, in line with the assumptions and caveats presented by Kleisner, Mansour & Pauly (2014). For a conceptual diagram of the workflow and calculations of the aMTL, see Liang & Pauly (2017).

To unmask the potential effects of the geographic expansion of fisheries on the MTC, we separated the fisheries catch based on species’ habitat classification (i.e., milieu) and also by the fishing gear used in order to re-calculate MTC trends for each group and gear type (Leitão et al., 2018). Since the overall MTC trend can be masked by the dominance of few (deep, cold-water) species, we separated catch by habitat associations and gear used, in an attempt to reduce the overall effect of dominant species on possible MTC trends of warming within different sectors of New Zealand fisheries. Species’ specific milieu identifications were done according to FishBase (Froese & Pauly, 2021, Table 1), while 23 distinct fishing gear classifications included in the catch data were aggregated into 12 general groups (Table S2). For the complete list of species caught by each gear type, see Table S3. We then calculate the MTC by gear, as the type of gear utilized will vary by the habitat exploited and thus determine the species composition of catch (Leitão et al., 2018). Similarly, we calculated MTC by milieu classification in order to group species based on similar environmental conditions and habitats utilized (Stephenson et al., 2020), and thus, distinguish between different habitats, geographic areas and species assemblages exploited throughout the period of fisheries expansion in the area. We also computed Kendall’s τ correlation between MTC values for each gear and milieu group and SSTA as well as the “full” MTC trend calculated for all analyzed species (n = 102).

Sensitivity analyses were completed in order to address the validity of the calculated metrics using the reconstructed Sea Around Us catch dataset. First, we computed the MTC, MTL, FiB and aMTL metrics using officially reported catch data from the FAO (2020). This dataset was composed of 42 species that were: both present in our calculations and also those of Durante, Beentjes & Wing (2020), had available trait information, and whose catch was reported at the species level (Table 1, Fig. 2C). Since gear information was not available in the FAO metadata, we thereafter calculated MTC by species’ milieu only. All species contained in this subset were QMS species, and represent 30% of the full catch data and 19% of all species/groups present in the FAO catch dataset. Second, we reduced the Sea Around Us dataset to those species contained in the FAO dataset (n = 42) and re-ran analyses. These 42 species represented 95% of the total analyzed Sea Around Us catch data (Fig. 2C).

Results

Across the study area and period, SSTA displayed a weak positive trend, increasing 0.04 °C per decade from 1950–2019 (Fig. 3A), with no significant breakpoint detected from segmented regression. When calculating the MTC of fisheries catch data, a segmented regression identified a significant breakpoint at 1986–whereby the MTC decreased 2.49 °C per decade from 1950–1986, while the MTC increased 0.44 °C per decade thereafter, i.e., from 1986–2019 (Table 2, Fig. 3B). Trends in MTL displayed a significant breakpoint at 1994, with the MTL increasing 0.15 per decade from 1950–1994, and decreasing from 1994–2019 (second slope = −0.03 per decade, Table 2, Fig. 3C). Results from the FiB index also reveal an overall positive trend, as the index increased 0.52 per decade until a detected breakpoint in 1999, while the subsequent slope decreased 0.11 per decade from 1999–2019 (Table 2, Fig. 3D). Thus, with modest levels of ocean warming across the study area, the MTC decreased sharply from 1950–1986, and has since gradually increased, whilst both the MTL and FiB indices increased sharply until the 1990s, and thereafter decreased gradually.

Figure 3 Results of calculated indices.

Trends for the various indices calculated across the New Zealand Exclusive Economic Zone (EEZ) and for fisheries catch from 1950–2019, including: (A) sea surface temperature anomaly (SSTA, °C), and its linear regression slope (red) and 95% confidence interval (grey band) (regression slope equation: y = −9.3 + 0.004x, Adj. R2 = 0.11, p-value = 0.002), (B) the mean temperature of the catch (MTC, °C), (C) the mean trophic level (MTL), and (D) the Fishing-in-Balance (FiB) index, including their segmented regression slopes. (E) Results from calculating the adapted Mean Trophic Level (aMTL), including the three identified stock assemblages (green, blue and red lines), as well as the years identified for node expansions (points) to include new stock assemblages (dashed lines).

Table 2 Results from simple linear regression and segmented linear regression for analyses of New Zealand’s fisheries catch data.

Index	Years	Decadal rate of change	Adj. R2	p-value	Breakpoint ± SE	
MTC	1950–2019	−1.14	0.62	<0.01		
1950–1986	−2.49	0.88	<0.01	1986 ± 1.7	
1986–2019	0.44				
MTL	1950–2019	0.09	0.74	<0.01		
1950–1994	0.15	0.86	<0.01	1994 ± 2.8	
1994–2019	−0.03				
FiB	1950–2019	0.29	0.74	<0.01		
1950–1999	0.52	0.96	<0.01	1999 ± 1.9	
1999–2019	−0.11				
Note:

Results from simple linear regression and segmented linear regression for the Mean Temperature of the Catch (MTC, °C), the Mean Trophic Level (MTL), and the Fishing-in-Balance Index (FiB) for New Zealand fisheries catch data from 1950–2019. For each index is listed the time period to which the regression applies, the decadal rate of change of the given index, the Adjusted R2 value, the p-value, as well as the year (±standard error SE) of a segmented regression breakpoint.

When calculating the aMTL, three distinct stock assemblages were identified when correcting for the geographic expansion of fisheries across New Zealand’s EEZ. In line with assumptions of the aMTL (Kleisner, Mansour & Pauly, 2014), the first stock assemblage includes the nearshore fisheries (Fig. 3E, green line), which maintained a stable aMTL from 1950–1964, thereafter decreasing until the 1990s. A second stock assemblage was identified in 1965 (Fig. 3E, blue line), indicating the first major geographic expansion of fisheries. In this case, fisheries started catching higher trophic level species compared to the initial nearshore assemblage, and aMTL also subsequently declined until the 1990s (Fig. 3E, blue line). The third stock assemblage was identified in 1969, which has since caught the highest trophic levels in New Zealand fisheries (aMTL values > 4) from 1969–2019 (Fig. 3E, red line).

Results from the aMTL confirmed the geographic expansion of New Zealand fisheries offshore throughout the study period. In turn, we separated fisheries catch data between species’ milieu and between fishing gear groups to re-calculate MTC trends. Results reveal a general pattern of MTC cooling for both bottom-associated species and fishing gears, with few groups displaying little to no MTC trend across the study period (Tables 3, 4, Figs. 4, 5). We do not report results for fishing gear groups ‘pots or traps’, ‘mixed gear’, ‘pelagic trawl’ or ‘pole and line’ as there was insufficient data to produce linear regression trends. The MTC for benthopelagic and demersal species displays strong reductions until 1979 and 1995, respectively (Table 3, Figs. 4C, 4D), with similar negative trends for bottom trawl (breakpoint: 1987), gillnet (breakpoint: 1999), longline (breakpoint: 1969), small scale (breakpoint: 1989) and unknown (breakpoint: 1979) fishing gear groups (Table 4, Fig. 5). Conversely, strong positive trends in MTC were observed for pelagic-oceanic species following 1965 (Table 3, Fig. 4F) and for purse seine fisheries (Fig. 5F) across the study period, as well as the ‘other’ fishing gear group after 1964 (Table 4, Fig. 5E). Overall, these results reflect the general trend of MTC cooling for bottom-associated species and fisheries, but also reveal warming MTC trends for pelagic and oceanic species across the study area.

Table 3 Regression and correlation results for the MTC between species milieu.

Milieu	Years	Decadal rate of change	Adj. R2	p-value	Breakpoint ± SE	Kendall’s τ corr. with SSTA (p-val)	Kendall’s τ corr. with MTC (p-val)	
Bathydemersal	1950–2019	0.14	0.78	<0.01		0.07 (p = 0.41)	−0.43 (p =< 0.01)	
Bathypelagic	1975–2019	0.32	0.58	<0.01		0.35 (p =< 0.01)	0.14 (p = 0.17)	
1975–1980	3.14	0.85	<0.01	1980 ± 0.59			
1980–2019	0.19						
Benthopelagic	1950–2019	−0.98	0.60	<0.01		−0.11 (p = 0.18)	0.70 (p =< 0.01)	
1950–1979	−2.54	0.84	<0.01	1979 ± 1.97			
1979–2019	−0.01						
Demersal	1950–2019	−0.43	0.50	<0.01		−0.04 (p = 0.64)	0.57 (p =< 0.01)	
1950–1995	−0.82	0.73	<0.01	1995 ± 2.31			
1995–2019	0.55						
Pelagic-neritic	1950–2019	−0.04	<0.01	0.30		0.01 (p = 0.86)	0.19 (p = 0.02)	
1950–1980	−0.39	0.22	<0.01	1980 ± 4.28			
1980–2019	0.20						
Pelagic-oceanic	1950–2019	1.46	0.45	<0.01		0.12 (p = 0.13)	−0.17 (p = 0.04)	
1950–1965	−4.01	0.62	<0.01	1965 ± 2.39			
1965–2019	2.24						
Reef-associated	1950–2019	<0.01	<0.01	0.58		0.03 (p = 0.74)	−0.33 (p =< 0.01)	
Note:

Results from simple linear regression and segmented linear regression for the Mean Temperature of the Catch (MTC, °C) of New Zealand fisheries data, from 1950–2019, separated by species’ milieu. Included is the time period to which the regression applies to, the decadal rate of change, the Adjusted R2 value, the p-value, as well as the location of a segmented regression breakpoint (±standard error SE). Also listed is each MTC’s Kendall’s τ correlation value (and p-value) with Sea Surface Temperature Anomaly (SSTA) and the full Sea Around Us MTC trend.

Table 4 Regression and correlation results for the MTC between fishing gear used.

Fishing gear	Years	Decadal rate of change	Adj. R2	p-value	Breakpoint ± SE	Kendall’s τ corr. with SSTA (p-val)	Kendall’s τ corr. with MTC (p-val)	
Bottom trawl	1951–2019	−1.42	0.71	<0.01		−0.13 (p = 0.12)	0.75 (p =< 0.01)	
1951–1987	−2.87	0.92	<0.01	1987 ± 1.44			
1987–2019	0.24						
Gillnet	1951–2019	−0.92	0.78	<0.01		−0.16 (p = 0.06)	0.49 (p =< 0.01)	
1951–1999	−1.19	0.84	<0.01	1999 ± 3.44			
1999–2019	0.12						
Hand lines	1951–2019	−0.20	0.01	0.20		0.04 (p = 0.65)	0.32 (p =< 0.01)	
Longline	1950–2019	−0.33	0.11	<0.01		0.02 (p = 0.80)	0.15 (p = 0.06)	
1950–1969	−3.05	0.51	<0.01	1969 ± 2.10			
1969–2019	0.30						
Other	1955–2018	0.93	0.36	<0.01		0.04 (p = 0.63)	−0.15 (p = 0.09)	
1955–1964	8.49	0.58	<0.01	1964 ± 1.78			
1964–2018	0.45						
Purse seine	1951–2019	0.86	0.12	<0.01		−0.02 (p = 0.80)	−0.01 (p = 0.95)	
Small scale	1950–2019	−1.11	0.66	<0.01		−0.12 (p = 0.13)	0.84 (p =< 0.01)	
1950–1989	−2.17	0.87	<0.01	1989 ± 1.88			
1989–2019	0.39						
Unknown	1951–2019	−0.64	0.26	<0.01		0.01 (p = 0.88)	0.49 (p =< 0.01)	
1951–1979	−3.07	0.81	<0.01	1979 ± 1.38			
1979–2019	0.79						
Note:

Results from simple linear regression and segmented linear regression for the Mean Temperature of the Catch (MTC, °C) of New Zealand fisheries data, from 1950–2019, separated by fishing gear. Included is the time period to which the regression applies to, the decadal rate of change, the Adjusted R2 value, the p-value, as well as the location of a segmented regression breakpoint (±standard error SE). Also listed is each MTC’s Kendall’s τ correlation value (and p-value) with Sea Surface Temperature Anomaly (SSTA) and the full Sea Around Us MTC trend.

Figure 4 MTC results per species’ milieu.

The mean temperature of the catch (MTC, °C), of New Zealand’s fisheries from 1950–2019, and their regression lines, separated by species’ milieu, including: (A) bathydemersal, (B) bathypelagic, (C) benthopelagic, (D) demersal, (E) pelagic-neritic, (F) pelagic-oceanic and (G) reef-associated. Listed in panels is the number of species included in each milieu (n), as well as the mean temperature preference (°C ± SE) of all included species.

Figure 5 MTC results per fishing gear group.

The mean temperature of the catch (MTC, °C), of New Zealand’s fisheries from 1950–2019, and their regression lines, separated by fishing gear, including: (A) bottom trawl, (B) gillnet, (C) hand lines, (D) longline, (E) other, (F) purse seine, (G) small scale and (H) unknown. Each panel lists the number of species included in each fishing gear’s catch (n), as well as the mean temperature preference (°C ± SE) of all included species in that gear group.

Results from Kendall’s τ correlation found only the bathypelagic MTC trend to be significantly positively correlated with SSTA trends (Table 3), while the rest of the milieu and fishing gear MTCs were not correlated, including the primary MTC trend. Correlations with the primary dataset (n = 102) MTC trend were strongest for those milieu and gears that had strong MTC reductions, including benthopelagic and demersal species, as well as bottom trawl and small-scale gears (Table 4). These results show that milieu- or gear-specific MTC trends were not strongly associated with low levels of ocean warming across New Zealand waters, while the primary MTC trend is correlated with bottom associated species- and fishing gear trends of MTC cooling.

Sensitivity analyses calculating all of the aforementioned metrics using FAO catch data (n = 42) and a reduced Sea Around Us dataset (n = 42) reveal similar results from the primary Sea Around Us dataset (n = 102) analyzed (Tables S4, S5, Figs. S1–S4). This includes the identification of three stock assemblages via the aMTL, with the first two assemblages displaying fishing down trends, although the years of identified expansion into new stock assemblages are 6 and 8 years later, respectively, within the FAO dataset (Fig. S1D). Milieu- and gear-specific MTCs also reveal similar trends, with negative slopes in bottom-associated fisheries and species. For the pelagic-oceanic milieu MTC, FAO data displayed strong warming until 2009, thereafter decreasing (Fig. S2F), while the Sea Around Us data subset displayed a consistent positive slope (Fig. S4F). Although for purse seine fisheries, strong trends of MTC warming were not present within the Sea Around Us subset (Fig. S4L). Overall, sensitivity analyses support the validity of our findings from the primary Sea Around Us dataset.

Discussion

Our results have identified and unmasked aspects of MTC warming during the geographic expansion of New Zealand fisheries catch into offshore waters from 1950–2019. By separating fisheries catch between species’ habitat and by fishing gear, we observed warming trends of pelagic-oceanic species across the study period. Results from sensitivity analyses confirmed a warming trend of pelagic-oceanic species, although FAO data showed this trend to occur from 1950–2009, thereafter reversing (Fig. S2F). As well, sensitivity analyses using the reduced Sea Around Us dataset removed strong trends of warming in purse seine fisheries (Fig. S4L), and thus, must be considered when interpreting the primary results of purse seine MTC warming. Nonetheless, overall results reflect the increasing proportion of pelagic-oceanic species of warmer thermal affinity caught in New Zealand from 1950 until at least 2009. This trend was otherwise masked in the full MTC by the geographic expansion and deepening of fishing operations (Fig. 3B). Strong correlations between bottom-associated habitat- and gear-specific MTC trends and the full, primary MTC suggests that cooling resulted from deepening of New Zealand’s catch. This similarly masked result in New Zealand waters was reported by Cheung, Watson & Pauly (2013) during the initial application of the MTC.

Following the calculation of the primary, masked MTC, in a step-wise process we calculated the MTL of the same catch data. These results were in line with those reported by Durante, Beentjes & Wing (2020) who also analyzed New Zealand fisheries, displaying a strong positive trend of MTL values until the early 2000s, and thereafter decreasing. This increasing trend of MTL, that indicated harvesting of higher trophic level species, corresponds to the period of geographic expansion and deepening of fisheries into colder waters in the area (Leitão, 2015), as new habitats and species (i.e., of higher trophic levels) were gradually included into New Zealand fisheries (Durante, Beentjes & Wing, 2020). Thereafter, incorporating the MTL into FiB and aMTL indices, we identified two discrete periods of fisheries expansion, with the first occurring in 1965, and the second in 1969.

The geographic expansion of New Zealand’s fisheries tracks the modernization of the country’s fisheries technology as well as policy. By the 1950s and 1960s, technologically-accessible nearshore fisheries were approaching saturation, and began to show signs of overexploitation (Stock Assemblage 1, Fig. 3E), coinciding with widescale deregulation measures for nearshore fisheries in 1963 (Clark, Major & Mollett, 1988; Pinkerton, 2017; Durante, Beentjes & Wing, 2020). Deregulation facilitated the rapid development of a domestic fisheries industry within the newly established 12 nautical mile Territorial Sea in 1965 (Bradstock, 1979; Clark, Major & Mollett, 1988; Fig. 1). This led to an expansion of fishing effort in shelf waters (Jackson et al., 2001; Kleisner, Mansour & Pauly, 2014), and coincided with our identification of the geographic expansion to include the second stock assemblage in 1965 (Fig. 3E). As we assume that this expansion occurred within shelf waters (sensu Kleisner, Mansour & Pauly, 2014), these two stock assemblages are parallel geographically (along New Zealand’s shelf) and thus maintain similar species assemblages (in contrast to the offshore Stock Assemblage 3). In turn, our results show similar trends of fishing down to have occurred in Stock Assemblages 1 and 2, which represents the full exploitation of species and thus the range of trophic levels available to fisheries within shelf waters during this period of fisheries expansion in New Zealand. A similar trend was observed between the first and second stock assemblage identified during the analysis of geographic expansion of fisheries in Indian waters (Kleisner, Mansour & Pauly, 2014).

As shelf-water fisheries continued to develop throughout the 1960s, high trophic level species were being depleted, thus Stock Assemblages 1 and 2 began to show signs of fishing down (Fig. 3E). This trend was revealed by the application of the FiB and aMTL indices, and was otherwise masked by the calculation of MTL of fisheries catch across the New Zealand EEZ (Fig. 3C, Durante, Beentjes & Wing, 2020). Similar results from Leitão (2015) reported signals of overfishing in nearshore stocks during a period of fisheries expansion, and thus support our results, as well as the assumptions of aMTL analyses, in that persistent fishing down in nearshore and shelf stock assemblages (i.e., Stock Assemblage 1 and 2) incentivizes expansion into previously unexploited stock assemblages (i.e., Stock Assemblage 3) (Kleisner, Mansour & Pauly, 2014). Concurrently, through further economic incentives including subsidies for new vessels, new processing plants, and an expanded list of exploitable species, New Zealand’s fisheries fleets expanded further offshore (Johnson & Haworth, 2004; Durante, Beentjes & Wing, 2020), coinciding with the expansion of fisheries into the third identified stock assemblage in 1969 (Fig. 3E). This third stock assemblage has maintained the highest mean trophic levels of catch across the time series (Fig. 3E), likely due to consistently high catches of the commercially-important hoki (Macruronus novaezelandiae, TL = 4.5, Table 1) since the early 1970s.

By completing sensitivity analyses with the reduced Sea Around Us catch data (n = 42) and the FAO catch data (n = 42), our results of fishing down within nearshore stock assemblages was confirmed throughout the period of fisheries expansion in New Zealand waters. The identification of node expansion into new stock assemblages occurred later within the aMTL from FAO data. This is expected, as FAO reported catch quantities are lower than those for the reconstructed Sea Around Us catch, including but not limited to the period from the 1950s to 1970s (Fig. 2C). Moreover, it is documented that the identification of node expansion can occur later if catch increases more gradually (Kleisner, Mansour & Pauly, 2014), as is the case for the analyzed FAO catch data.

By separating and re-calculating the MTC between species’ milieu and fishing gear type, MTC trends are in line with results reported via the aMTL. Several fisheries with bottom-associated gear or species, including deep-water species, displayed negative MTC trends from the mid- to late 1960s until the 1980s and 1990s (Figs. 4, 5). This cooling of the MTC coincides with the expansion of fishing operations to include Stock Assemblage 3 from 1969 onwards. As gear development promoted further expansion into deeper and cooler waters, new catches of barracouta (Thyrsites atun), southern blue whiting (Micromesistius australis) and hoki drove down MTC trends (Figs. 6A, 6B). By the 1980s, catches of hoki dominated not only gear- and habitat-specific fisheries catch, but also overall catch in New Zealand waters. This led to the levelling off of both hoki-dominated gear- and habitat-specific MTCs (e.g., Figs. 4C, 5A) and also the general trend of New Zealand’s MTC (Fig. 3B).

Figure 6 Catch trends for most abundant bottom-associated and pelagic-associated species.

Species with the highest overall catches (tons × 103) across the study period for: (A) benthopelagic species, (B) species caught in bottom trawl fisheries, (C) pelagic-oceanic species and (D) species caught in purse seine fisheries.

For pelagic-oceanic species and pelagic purse seine fisheries, trends of MTC warming were driven by the increasing catch of the subtropical skipjack tuna (Katsuwonus pelamis) and blue mackerel (Scomber australasicus), paired with the decreasing catch of the temperate southern bluefin tuna (Thunnus maccoyii, Figs. 6C, 6D). These results are in line with observations of pelagic species undergoing relatively rapid shifts in geographic distribution with oceanographic changes (Champion, Brodie & Coleman, 2021), albeit the changes in SSTA of the present study area are modest. Such changes in the composition of pelagic-oceanic species present in fisheries catch may be driven by the adult fishes’ ability to shift synchronously with their pelagic habitat to track local climate velocities (Pinsky, Selden & Kitchel, 2020; García Molinos et al., 2022). Moreover, these results suggest that the observed levels of ocean warming may favor the increased abundance of subtropical, smaller-bodied skipjack tuna and blue mackerel vs the temperate and larger-bodied southern bluefin tuna. In line with these results, and in recognition of the complexities of variable shifts in geographic distribution between different species (Dunn et al., 2022), we recommend that further research investigates the variability of shifts in geographic distribution of teleost fish species in line with ocean warming across New Zealand waters.

Our results of the positive trend in SSTA are in line with observations of the gradual warming of New Zealand waters. Overall, ocean warming within New Zealand’s EEZ has exhibited relatively weak trends and significant interannual variability due to the dynamic influences of the warm Subtropical Front (STF) from the north, north-west and the cold Subantarctic Front (SAF) from the south (Shears & Bowen, 2017; Sutton & Bowen, 2019). Management must consider the additive or synergistic effects of both fishing pressure as well as ocean-warming-induced changes in productivity and geographic distribution of fish populations. For example, fishing effort may be reduced on stocks that are overexploited and/or are shifting the trailing edge of their geographic distribution away from historically fished areas as they track their thermal affinity (Szuwalski & Hollowed, 2016; Gaines et al., 2018; Lamine et al., 2022). In the present study, this may apply to the cold-water southern bluefin tuna, which displays strong reductions in overall catch from the 1970s onward (Fig. 6C). This may also be relevant as other commercially-important species such as hoki are predicted to shift their geographic distribution polewards with future ocean warming (Dunn et al., 2022). Conversely, the leading edge of pelagic, warm-water skipjack tuna and blue mackerel may be penetrating deeper into New Zealand’s EEZ, thus facilitating higher catches. The joint application of the present indices can help inform such management decisions.

Interpretation of trends in the MTC should also consider historical fishing pressure and management within this particular system. Demersal or reef-associated species like the Australasian snapper Pagrus auratus were already displaying signs of overexploitation by the 1970s (Durante, Beentjes & Wing, 2020), as confirmed by our calculation of the nearshore and shelf aMTLs (Stock Assemblage 1 and 2, Fig. 3E). As a result, New Zealand implemented their quota management system (QMS) in 1986 (Clark, Major & Mollett, 1988), and since then, overall fisheries catches have decreased (Fig. 2A), and that of Australasian snapper has remained steady but well-below its maximum catch values seen in the 1960s and 1970s (Supplemental Files). Conversely, since the 1980s, skipjack tuna and blue mackerel have displayed oscillating but gradually increasing catches (Figs. 6C, 6D). This may be a result of the then newly-implemented QMS, the technological and geographic expansion of fisheries in New Zealand waters, as well as trends in ocean warming and the increased abundance of pelagic, warm-water fishes in the area. We therefore recommend that interpretation of the present MTC warming trend considers both the thermal affinity tracking of pelagic species with ocean warming, as well as fisheries management in New Zealand adapting to stock changes.

Conclusions

Overall, our results build on previous research of the MTL to correct for the geographic expansion of fisheries within the New Zealand EEZ from the 1950s onward. Using the FiB and aMTL indices, we identified trends of fishing down trophic levels in nearshore fishing stocks, a trend that was otherwise masked within the full calculation of New Zealand’s MTL (Durante, Beentjes & Wing, 2020). Thereafter, by separating catch between species’ milieu and fishing gear groups, increasing catches of subtropical skipjack tuna and blue mackerel revealed positive MTC trends for pelagic species and fishing gear. These trends were also otherwise masked by the offshore expansion and deepening (into colder depths) of fishing operations and catch, as species such as hoki came to dominate overall catches across the same time period. These results highlight the pervasiveness of fishing down food webs in New Zealand fisheries throughout a period of technological development and fisheries geographic expansion. These results also reveal how fisheries catch composition may change due to the responsiveness of pelagic, mobile species to track local climate velocities with ocean warming.

Supplemental Information

Supplemental Information 1 Sea Around Us catch data for 102 analyzed species.

Annual catch values (tons) of analyzed species within the Sea Around Us dataset (n = 102).

Click here for additional data file.

Supplemental Information 2 Official FAO catch data analyzed for 42 species.

Annual catch values (tons) of analyzed species within the FAO dataset (n = 42).

Click here for additional data file.

Supplemental Information 3 Filtering data from the Sea Around Us fisheries catch dataset.

The species and higher taxonomic groups recorded in the full Sea Around Us New Zealand EEZ catch dataset, from 1950–2019, that were excluded from analyses in the present study. Listed is the species/group name, as well as its taxonomic classification.

Click here for additional data file.

Supplemental Information 4 Fishing gear groupings.

The gear used, from 1950–2019, within New Zealand’s EEZ fisheries catch data, as reported by the Sea Around Us. Right: The broader fisheries gear groupings used in analyses for all listed gear within the fisheries catch data.

Click here for additional data file.

Supplemental Information 5 Recorded species per fishing gear group.

The list of study species recorded as catch between the distinct gear groups. Note: Several species are captured in more than one gear.

Click here for additional data file.

Supplemental Information 6 Results from simple linear regression and segmented linear regression for sensitivity analyses of New Zealand fisheries catch data.

Results from simple linear regression and segmented linear regression for the Mean Temperature of the Catch (MTC, °C), the Mean Trophic Level (MTL), and the Fishing-in-Balance Index (FiB) for New Zealand fisheries catch data from FAO catch statistics and the reduced Sea Around Us dataset (n = 42). For each index is listed the time period to which the regression applies, the decadal rate of change of the given index, the Adjusted R2 value, the p-value, as well as the year (± standard error SE) of a segmented regression breakpoint.

Click here for additional data file.

Supplemental Information 7 Regression and correlation results for the MTC between species’ milieu and fishing gear groups for sensitivity analyses.

Results from simple linear regression and segmented linear regression for the Mean Temperature of the Catch (MTC, °C) of New Zealand fisheries data, separated by species’ milieu and by fishing gear between FAO catch data and the reduced Sea Around Us data (n = 42). The FAO data only analyzed species’ milieu, while the reduced Sea Around Us data analyzed milieu and gear. Included is the time period to which the regression applies to, the decadal rate of change, the Adjusted R2 value, the p-value, as well as the location of a segmented regression breakpoint (± standard error SE). Also listed is each MTC’s Kendall’s τ correlation value (and p-value) with Sea Surface Temperature Anomaly (SSTA) and the full Sea Around Us MTC trend.

Click here for additional data file.

Supplemental Information 8 Results of calculated indices from FAO catch data for sensitivity analyses.

Trends for the various indices calculated across the New Zealand Exclusive Economic Zone (EEZ) and for FAO fisheries catch data (n = 42) from 1950–2019, including: (a) the mean temperature of the catch (MTC, °C), (b) the mean trophic level (MTL), and (c) the Fishing-in-Balance (FiB) index, including their segmented regression slopes. (d) Results from calculating the adapted Mean Trophic Level (aMTL), including the three identified stock assemblages (green, blue and red lines), as well as the years identified for node expansions (open circles) to include new stock assemblages (dashed lines). The first identified expansion from Stock Assemblage 1 to Stock Assemblage 2 occurred in 1971, while the second expansion from Stock Assemblage 2 to Stock Assemblage 3 occurred in 1977.

Click here for additional data file.

Supplemental Information 9 FAO MTC results per species’ milieu from sensitivity analyses.

The mean temperature of the catch (MTC, °C), of New Zealand’s fisheries from 1950–2019, separated by species’ milieu, from FAO catch data (n=42). This includes: (a) bathydemersal, (b) bathypelagic, (c) benthopelagic, (d) demersal, (e) pelagic-neritic, and (f) pelagic-oceanic species. Listed in panels is the number of species included in each milieu (n), as well as the mean temperature preference (°C, ± SE) of all included species.

Click here for additional data file.

Supplemental Information 10 Results of calculated indices from reduced Sea Around Us catch data for sensitivity analyses.

Trends for the various indices calculated across the New Zealand Exclusive Economic Zone (EEZ) from 1950-2019 using the reduced Sea Around Us (n = 42) dataset, including: (a) the mean temperature of the catch (MTC, °C), (b) the mean trophic level (MTL), and (c) the Fishing-in-Balance (FiB) index, including their segmented regression slopes. (d) Results from calculating the adapted Mean Trophic Level (aMTL), including the three identified stock assemblages (green, blue and red lines), as well as the years identified for node expansions (open circles) to include new stock assemblages (dashed lines). The first identified expansion from Stock Assemblage 1 to Stock Assemblage 2 occurred in 1965, while the second expansion from Stock Assemblage 2 to Stock Assemblage 3 occurred in 1969.

Click here for additional data file.

Supplemental Information 11 MTC results per species’ milieu and fishing gear group from reduced Sea Around Us dataset for sensitivity analyses.

The mean temperature of the catch (MTC, °C), of New Zealand’s fisheries from 1950–2019, separated by species’ milieu, from the reduced (n = 42) Sea Around Us dataset. This includes: (a) bathydemersal, (b) bathypelagic, (c) benthopelagic, (d) demersal, (e) pelagic-neritic, and (f) pelagic-oceanic species, plus (g) bottom trawl, (h) gillnet, (i) handline, (j) longline, (k) other, (l) purse seine, (m) small scale, and (n) unknown fishing gear groups. Listed in panels is the number of species included in each group (n), as well as the mean temperature preference (°C, ± SE) of all included species.

Click here for additional data file.

Additional Information and Declarations

Competing Interests

Author Contributions

Data Availability

Mark John Costello is an Academic Editor for PeerJ.

Charles Patrick Lavin conceived and designed the experiments, performed the experiments, analyzed the data, prepared figures and/or tables, authored or reviewed drafts of the article, and approved the final draft.

Daniel Pauly conceived and designed the experiments, authored or reviewed drafts of the article, and approved the final draft.

Donna Dimarchopoulou conceived and designed the experiments, authored or reviewed drafts of the article, and approved the final draft.

Cui Liang conceived and designed the experiments, performed the experiments, analyzed the data, authored or reviewed drafts of the article, and approved the final draft.

Mark John Costello conceived and designed the experiments, authored or reviewed drafts of the article, and approved the final draft.

The following information was supplied regarding data availability:

The data utilized in the present study, as well as code to perform analyses, is available from Zenodo: Charles P. Lavin. (2023). Code and data for: Fishery catch is affected by geographic expansion, fishing down food webs and climate change in Aotearoa, New Zealand. Zenodo. https://doi.org/10.5281/zenodo.8207722

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
