# Peer review of "Fishery catch is affected by geographic expansion, fishing down food webs and climate change in Aotearoa, New Zealand"

_PeerJ, doi:10.7717/peerj.16070_

## Round 0.1 · original submission · Major Revisions

Three reviewers have provided comments. One of them found severe problems with the validity of the input catch data, affecting the derived metrics (MTC, MTL, FiB, and aMTL). Another reviewer questions the objective of the work in which it has to be clarified if whether changes in catch composition in NZ waters are driven by ocean warming and/or by the geographical expansion of fisheries due to fishing down the food web. The other reviewer questions why the basket shark was included since this is an incidental catch.

Given the importance of time series as a surrogate of the condition of populations affected by fisheries, I found your work interesting, but to proceed with another round of reviews I strongly recommend removing unnecessary data and carefully following the recommendations provided by reviewers. You need to clearly establish the overall question and then proceed to respond to that question based on solid time series (see what one of the reviewers said about catch histories for black oreo (Allocyttus niger) and smooth oreo (Pseudocyttus maculatus), which are wrong.

Please, respond to those recommendations and include a clean, new version of your manuscript no later than the provided deadline. Thank you.

·

Basic reporting

This paper uses regressions fitted to derived indices in an attempt to investigate changes in trophic level and fish distribution in relation to catch and climate. The authors used publicly available data sources to describe catch (Sea Around Us database), mean trophic level (FishBase) and mean temperature preference (distribution modelling of Cheung et al. 2013). The study area was the EEZ of Aotearoa New Zealand and analyses were based on 109 species.

The topic of the interaction between fishing and changing climate is of great interest. The paper is well written although the outcomes seem somewhat pre-determined – the authors “expect fishing down to occur” and “expect signals of warming in the MTC” (lines 106–109) rather than testing null hypotheses expressed with a more neutral wording e.g., “investigate whether fishing down occurred”.

Experimental design

My major concern is in the validity of the input catch data, which then throws doubts on the derived metrics (MTC, MTL, FiB, and aMTL). I am familiar with the species and catch histories of many of the 109 species and by looking through the catch histories in Figs S1-S107 (I think two species are not plotted as they have fewer than 2 years of catch data; lines 145-146), I have the following concerns:

1. The Sea Around Us (SAU) dataset reconstructs catch histories based on “reported statistics as well as reconstructed reports of discards and bycatch” (lines 130–131). The reconstruction process involves assumptions and has the effect of inflating estimated catches from those reported by official NZ government statistics (given in Fisheries New Zealand (2022). Fisheries Assessment Plenary, May 2022: stock assessments and stock status. Compiled by the Fisheries Science Team, Fisheries New Zealand, Wellington, New Zealand. 1886 p.). I am not going to argue here which alternate catch history is more “correct” but suggest that a sensitivity on derived indices based on official catch histories would be informative to see how much the assumptions used to derive SAU data impacts results. For example, Durante et al. (2020) includes a good discussion of various catch histories (FAO/FNZ/SAO) and the data set they used for 67 species based on FNZ & FAO data might make a useful alternative for sensitivity. As indices are catch weighted, this may not make a material difference?

2. Catch histories for many of the rarer of the 109 species (those with estimated catches of less than 10 t per year in Table S4) are likely very poorly estimated as these species are often not landed and are only rarely observed or reported. Again this might not make a difference, but a sensitivity based on the more abundant (and hence more reliably reported) species would be informative (e.g., the subset of 67 species used by Durante et al. (2020) would likely still account for a very high proportion of catch).

3. Catch histories for black oreo (Allocyttus niger) and smooth oreo (Pseudocyttus maculatus) are wrong with only small catches reported. This is because the large catches of both species were usually reported under a combined oreo code.

4. Something is odd with catch history for tarakihi (Nemadactylus macropterus). Catches should be in maximum of ~4000 t?

5. Similarly wrong catch history for trevally (Pseudocaranx dentex). Catches should be in maximum of ~3000 t? And this looks identical to the plot for Nemadactylus macropterus?

6. Wrong catch history for hapuku (Polyprion oxygeneios) as catches were usually only reported to genus.

7. Wrong catch history for gemfish (Rexea solandri) with spike in only one year. Catches should have maximum of ~7000 t?

8. Catches reported for Trachurus declivis but not Trachurus novaezelandiae? Jack mackerel catches are usually only reported to genus but both species important (also T. murphyi).

9. No reported catch for john dory (Zeus faber) in about 2016–2017. Catches of 600–700 t were reported in this period.

Validity of the findings

I’m not sure how much any of the above errors influence the results and conclusions reported in the paper, but I was not willing to thoroughly evaluate the rest of the manuscript with this uncertainty. A quick glance at the estimated mean preference temperature and trophic level by species in Table S1 suggests that reasonable values were used for most species, although I am not convinced that inferences about distribution and effects of warming climate based on MTC are informative when many species occupy a broad temperature band and observed temperature changes around the NZ EEZ haven’t been large, particularly at depth.

Additional comments

The abstracts and conclusions both make relatively strong statements based on potentially (and in some cases demonstrably) erroneous data. I cannot recommendation publication in the current state. As well as correcting errors, an investigation of the sensitivity of derived metrics to the subset of species selected and their assumed catch histories would greatly improve this manuscript.

Reviewer 2 ·

Basic reporting

This is a interesting paper that brings novel perspective in use marine ecological indexes to understand trends due to fishery and climate.
I have little comments to the MS, check below (some are in the pdf)
- The stock assemblage relationships with marine trophic indexes were also addressed by Leitão et al., (2015) with the same approximately results ; the expansion of the fishery was due to the increase of the fishery to higher trophic levels. Maybe introduction and discussion benefits with this reference to support present findings (e.g. L244-249; L282) namely considering results of both studies and also evidences of changes in fish community structure in early 2000s due to fishery expansion.

- Time series techniques would allow to identify interactions between the estimated common trends ((a)MTL and MTC), explanatory variables and each individual function group (stock assemblages) trend (e.g check Zurr et al studies). Therefore, I felt that potentially data between different time series could be better compared if time series analyses were used. For instance, changes in both (a)MTL and MTC can be related with SST anomalies (explanatory variable) and also with Fig 5 time series. This would allow simplified interpretation of results and support discussion (see Leitão et al. 2015). As these as not tested it is difficult to understand trends relationships and so to support some discussion statements (e.g. L270-272 and other discussion sections) which become more speculative and base on authors trends perception/interpretation. Such analyses will help support discussion making it less speculative and supporting/linking results based on statistical analyses (for instance in what extend adapted and standard MTC are different or not along the time series?).

Zuur A.F., Frywe R.J., Jolliffe I.T., Dekker R. & Beukema J.J. (2003) Estimating common trends in multivariate time series
using dynamic factor analysis. Environmetrics 15, 665–668
Leitão F 2015. Landing profiles of Portuguese fisheries: assessing the state of stocks. Fisheries Management and Ecology 22 (2), 152-163

- I don’t understand why authors did not exclude Cetorhinus maximus (Basking shark) because this is potentially accidental catch of a species without any particular economic interest that potentially is not target by any fleet type so it can bias information on pelagic ocean species and also MTC and MTL? As any reason to keep it (check Fig. 6). As data comes from SAUs this can be result of the rebuild catch but even taught I believe it should be removed from analyses (at least stock assemblage and MTL/MTC relationships).

- Maybe the number of figures can be reduced to an excel spreadsheet with catch yearly data by species and or species can be grouped reducing substantially the supplements.

Experimental design

this paper does not deals with complex Exprimental Design; it is a determination of marine ecological indexes and trends analyses.
Time series techniques would allow to identify interactions between the estimated common trends ((a)MTL and MTC), explanatory variables and each individual function group (stock assemblages) trend (e.g check Zurr et al 2003 studies). Therefore, I felt that potentially data between different time series could be better compared if time series analyses were used. For instance, changes in both (a)MTL and MTC can be related with SST anomalies (explanatory variable) and also with Fig 5 time series. This would allow simplified interpretation of results and support discussion (see Leitão et al. 2015). As these as not tested it is difficult to understand trends relationships and so to support some discussion statements (e.g. L270-272 and other discussion sections) which become more speculative and base on authors trends perception/interpretation. Such analyses will help support discussion making it less speculative and supporting/linking results based on statistical analyses (for instance in what extend adapted and standard MTC are different or not along the time series?).

Alternative simple tests might include correlation analyses among time series in order to understand how different nature of the data relates among time series data

excluding above the paper cope wth peer j standards for ED

Validity of the findings

check comments on point 2 , experimental design.

Annotated reviews are not available for download in order to protect the identity of reviewers who chose to remain anonymous.

·

Basic reporting

I think this is a well-written and well-executed study.

Experimental design

What is the overall question of this paper? It took me a while to work it out, and I am still not quite sure if I got it right.
Is your study trying to address the question whether changes in catch composition in NZ waters are driven by ocean warming and/or by the geographical expansion of fisheries due to fishing down the foodweb?
If this is correct, it would be good to change the title to reflect the goal of your study and state it upfront in the abstract/introduction. Be very clear what the ‘masking effect of the geographic expansion of fisheries’ means.
I think the main points of your study are that you are extending Durante et al.’s (2020) study to include an adapted MTL to show that fishing down the foodweb has occurred concurrently with fishery expansion. By quantifying this effect, you are able to ‘unmask this effect on the trend of MTC’ (e.g. line 213) – which I think means changes in the composition of fish catch due to ocean warming.

Validity of the findings

Data and analysis appear robust.

Additional comments

Line 106: it would be good to clearly state that the aMTL quantifies the geographical expansion of fishing.
Line 129-132: the authors use data from the Sea Around Us database. Is there any specific reason why data is not directly requested from the Ministry of Primary Industries (MPI)? I routinely request data from MPI and it is usually really good and detailed data. An explanatory sentence might suffice.
Lines 133-159: I think the data is well explained and the exposition of the MTC and MTL appears technically sound.
Line 164: It would be good to provide a short, intuitive explanation here about what the FiB index is (the authors do so in detail in line 173).
Line 215: can you add a bit more detail on how you re-calculate the MTC trends? I struggled to understand this. Specifically, how does the quantification of the geographic expansion of fisheries inform the separation of fish catch based on habitat classification and gear type? It is probably relatively straight forward but a bit more intuition would help.
Line 267-387: the discussion is very long-winded. I commend the authors for putting their results into historical context, but I would like to see a more concise discussion of the results.

---

## Round 0.2 · Minor Revisions

Two reviewers provided their decision. One of them is still requiring to address some minor revision. Please, make the required change to Fig.2C and double check the sensitivity analyses of the other species.

·

Basic reporting

I am satisfied with the authors' responses to my concerns and those of the other reviewers. The language is much more neutral and balanced. Adding the sensitivities based on the subset of 42 species (with both FAO and SAU catch statistics) greatly strengthens the paper from the original submission.

Experimental design

Adding the sensitivities based on the subset of 42 species (with both FAO and SAU catch statistics) greatly strengthens the paper from the original submission. The authors have responded to my other main concern by removing species where I identified errors in catch histories, but I hope that there were no errors in other species which I did not detect!

Validity of the findings

In Fig 2c I don't understand the difference between the "analyzed SAU dataset" (solid line) and the "further reduced SAU dataset" (dotted line)? I had thought that analyzed dataset was for 102 species and reduced dataset for the sensitivity with 42 species? But if this was the case why wasn't solid line in 2c the same as species line in 2b? Please clarify

Additional comments

No additional comments

·

Basic reporting

no comment

Experimental design

no comment

Validity of the findings

no comment

Additional comments

no comment

---

## Round 0.3 · accepted · Accept

The manuscript is ready for publication.